# Broad clinical manifestations of polygenic risk for coronary artery disease in the Women's Health Initiative

Shoa L. Clarke [1,2,10], Matthew Parham[3,10], Joanna Lankester [1,2], Aladdin H. Shadyab[4], Simin Liu [5], Charles Kooperberg [6], JoAnn E. Manson[7], Catherine Tcheandjieu[8,9] & Themistocles L. Assimes [1,2,3✉]

## Abstract

**Background** The genetic basis for coronary artery disease (CAD) risk is highly complex. Genome-wide polygenic risk scores (PRS) can help to quantify that risk, but the broader impacts of polygenic risk for CAD are not well characterized.

**Methods** We measured polygenic risk for CAD using the meta genomic risk score, a previously validated genome-wide PRS, in a subset of genotyped participants from the Women's Health Initiative and applied a phenome-wide association study framework to assess associations between the PRS and a broad range of blood biomarkers, clinical measurements, and health outcomes.

**Results** Polygenic risk for CAD is associated with a variety of biomarkers, clinical measurements, behaviors, and diagnoses related to traditional risk factors, as well as risk-enhancing factors. Analysis of adjudicated outcomes shows a graded association between atherosclerosis related outcomes, with the highest odds ratios being observed for the most severe manifestations of CAD. We find associations between increased polygenic risk for CAD and decreased risk for incident breast and lung cancer, with replication of the breast cancer finding in an external cohort. Genetic correlation and two-sample Mendelian randomization suggest that breast cancer association is likely due to horizontal pleiotropy, while the association with lung cancer may be causal.

**Conclusion** Polygenic risk for CAD has broad clinical manifestations, reflected in biomarkers, clinical measurements, behaviors, and diagnoses. Some of these associations may represent direct pathways between genetic risk and CAD while others may reflect pleiotropic effects independent of CAD risk.

### Plain language summary

An emerging method for predicting heart disease risk uses personal genetic information. Genetic risk is estimated by searching a person's genome for DNA changes (genetic variants) that are associated with heart disease. One tool sums the information provided by more than 1 million genetic variants. We hypothesized that these variants may impact health outcomes beyond heart disease. We tested this hypothesis using data from the Women's Health Initiative, a long-term study of post-menopausal women. We found that genetic risk for heart disease associated with many health outcomes. Some are risk factors for heart disease (e.g., high blood pressure), some are related to heart disease (e.g., stroke), and some outcomes appear unrelated and represent new avenues for research (e.g., cancer). Genetic testing may be a valuable approach to risk assessment, but we are still learning the complex nature of these tests.

[1] VA Palo Alto Health Care System, Palo Alto, CA, USA. [2] Department of Medicine, Division of Cardiovascular Medicine, Stanford University School of Medicine, Stanford, CA, USA. [3] Department of Epidemiology and Population Health, Stanford University School of Medicine, Stanford, CA, USA. [4] Herbert Wertheim School of Public Health and Human Longevity Science, University of California, San Diego, La Jolla, CA, USA. [5] Department of Epidemiology, Brown University, Providence, RI, USA. [6] Fred Hutchinson Cancer Research Center, Seattle, WA, USA. [7] Department of Medicine, Brigham and Women's Hospital, Harvard Medical School, Boston, MA, USA. [8] Gladstone Institute, San Francisco, CA, USA. [9] Department of Epidemiology and Biostatistics, University of California San Francisco, San Francisco, CA, USA. [10] These authors contributed equally: Shoa L. Clarke, Matthew Parham. ✉email: tassimes@stanford.edu

Coronary artery disease (CAD) is a complex phenotype, and the genetic basis for CAD risk is similarly complex[1]. To date, >200 loci have been implicated in CAD risk through genome-wide association studies (GWAS)[2,3]. These loci interact through a diverse set of biological pathways, and many loci have no apparent relevance to traditional risk factors for CAD. Furthermore, genetic variants that associate with CAD also associate with other phenotypes, suggesting extensive underlying pleiotropy[2,3]. The complexity of genetic risk for CAD is further highlighted by recent advances in the construction of polygenic risk scores (PRS). Contemporary scores that incorporate variants across the whole genome, including variants outside of known CAD loci, outperform scores that are constructed only from variants at known CAD loci[4,5]. Studying such genome-wide PRS for CAD may allow for improved understanding of the genetic basis for CAD risk and new insights into the implications polygenic risk beyond CAD.

One approach to assessing the impact of polygenic risk for CAD has been to measure associations between a CAD PRS and biobank-derived phenotypes[6,7]. A primary advantage of this approach is the large number of participants in such biobanks. However, a limitation of this method is lack of precision for some outcomes, particularly those inferred from electronic health records. Further, biobank studies have typically combined prevalent and incident disease and may have limited follow-up after enrollment. Thus, a complementary approach to biobank analyses is to examine well-phenotyped longitudinal cohorts.

Here, we seek to identify traits and outcomes associated with polygenic risk for CAD by taking advantage of the high-quality data collected as part of the Women's Health Initiative (WHI). We aggregate data collected over ~25 years as part of either clinical trials or the observational study within WHI. We thus measure the association between polygenic risk for CAD and blood biomarkers, clinical measurements, clinical risk scores/questionnaires, self-reported medical history, and incident adjudicated outcomes related to cardiovascular disease, cancer, and death.

## Methods

**Study cohort.** The main study cohort was selected from WHI. The design and recruitment strategy for WHI has been previously described[8,9]. Briefly, postmenopausal women aged 50 to 79 years were enrolled at 40 sites across the United States from 1993 to 1998. Each participant was enrolled into either a clinical trial ($n = 68{,}132$) or an observational study ($n = 93{,}676$). Two successive extension studies continued follow-up of consenting participants from 2005 to 2010 and from 2010 to the present. A subset of participants who were primarily non-Hispanic white by self-report have been previously genotyped as part of 6 ancillary GWAS (Supplementary Table 1). Participants from those 6 GWAS were considered for inclusion in this study. Because currently validated genome-wide PRS were developed in European populations and do not transfer well to non-European populations, we did not include cohorts of primarily non-European genotyped participants in this study. Subjects with a known or likely history of atherosclerotic cardiovascular disease (ASCVD) at enrollment were excluded (Supplementary Table 2). We used the UK Biobank for replication of select results. The UK Biobank cohort consisted of unrelated post-menopausal women of European ancestry with no history of MI or stroke at enrollment.

**Genotyping and imputation.** Genotyping was performed with early versions of Affymetrix and Illumina gene chips for five of the GWAS cohorts contributing to this study. For these five studies, harmonization and imputation to the 1000 Genome reference panel was previously performed as part of the WHI GWAS Harmonization and Imputation Project (https://www.ncbi.nlm.nih.gov/projects/gap/cgi-bin/study.cgi?study_id=phs000746.v3.p3). Participants of the sixth study were genotyped with the Oncochip, and we imputed these data to the 1000 Genome reference panel using the Michigan Imputation Server[10].

**Main exposure.** We used metaGRS, a previously developed genome-wide PRS for CAD, to estimate each participant's genetic risk[5]. This score consists of ~1.7 million autosomal variants. Participants in our study cohort did not contribute to the GWAS used to construct this score. Each participant's total score was calculated using Plink 2.0, and raw scores were then scaled to mean 0 and standard deviation 1. This standardize score was used as the primary exposure.

**Phenotypes.** Quantitative measurements were largely collected at enrollment and included laboratory values, clinical measurements, and clinical scores. For the small number of lab measurements not collected at baseline, we used the earliest available measurement. Lab outliers were removed by excluding the top 1% of values for each biomarker. For clinical measurements such as blood pressure, the mean value was used if serial measurements were available within one research clinic visit. Self-reported medical history, medication usage, social/behavioral history, and family history was obtained through questionnaires collected primarily at enrollment but also during annual follow-up mailings. Adjudicated outcomes assessed in this study include incident cardiovascular diseases, incident cancers, and death. Annual questionnaires were completed by participants or their proxies in order to identify hospitalizations, and for each hospitalization, medical records were obtained and adjudicated by physicians using standardized criteria[11]. Deaths were further ascertained through the National Death Index. For UK Biobank analyses, cancer diagnoses were extracted from the UK cancer registry. For each cancer, only first diagnoses after enrollment were considered as incident cases, and subjects with prevalent disease at enrollment were excluded.

**Statistical analysis.** We selected the largest subset of subjects with similar inferred genetic ancestry using principal components analysis in order to limit confounding by population substructure. We used linear and logistic regression to estimate associations between each trait/outcome and the CAD PRS per standard deviation increase in the PRS. For each of the adjudicated outcome, we appropriately censored subjects at the end of the follow-up time period where formal adjudication ended for the outcome. For death outcomes, we used Cox analysis with time zero being the time of enrollment. For each cause of death that was examined, non-cases were censored at time of death from another cause or time of last follow-up if not deceased.

Each model was adjusted for age at enrollment (or age at time of measurement for lab values), study type (clinical trial versus observational study), and genotyping platform. Associations with lipid-related labs, diabetes-related labs, and for blood pressure were additionally adjusted for self-reported cholesterol medication use, diabetes medication use, and hypertension medication use respectively. All associations with lab values were also adjusted for the assay version if more than one assay was used. For the analysis of self-reported outcomes, we compared three associations. First, we performed logistic regression using the main study cohort, adjusting for age at enrollment, study type, and genotyping platform. Second, we added an additional binary covariate to adjust for presence or absence of CAD at the last

follow-up. Third, we analyzed the subset of participants with no CAD at follow-up ($n = 18,044$), adjusting for age at enrollment, study type, and genotyping platform. CAD at follow-up was determined using both self-report and adjudicated outcomes. Only outcomes with at least 100 cases among the CAD-free cohort were considered, resulting in a total of 128 self-reported qualitative variables. The logistic regression analysis of adjudicated cardiovascular outcomes and the Cox analysis of death outcomes were adjusted for smoking status, self-reported diabetes at baseline, systolic blood pressure, low-density lipoprotein cholesterol (LDL-C), and high-density lipoprotein cholesterol (HDL-C). For cancer outcomes, we assessed the association with and without adjustment for risk factors. The risk factor adjusted model include adjustment for smoking status, alcohol consumption, physical activity (MET-hours per week), Alternative Healthy Eating Index score, and body-mass index (BMI).

Using a phenome-wide association study framework, we consider statistical significance in three ways. Nominal significance is defined as a $p$-value $\leq 0.05$. Where indicated, we also identify associations that are significant by a false-discovery rate (FDR) $q$-value $\leq 0.05$, using the Benjamini and Hochberg method. Lastly, Bonferroni significance is defined as 0.05 divided by the number of association tests performed for the given analysis. For the association analysis of quantitative traits, Bonferroni significance was $p$-value $\leq 9.2 \times 10^{-5}$ (0.05/546). For the association analysis of incident cardiovascular diseases, Bonferroni significance was $p$-value $\leq 0.003$ (0.05/17).

We used published summary statistics from GWAS of CAD[2], breast cancer[12], and lung cancer[13] to estimate genetic correlations. For CAD, only variants with INFO score >0.9 were included. For breast and lung cancer, INFO score was not available, and thus only HapMap3 variants were included, as these variants are generally well imputed. We used *ldsc* (version 1.01) to perform genetic correlation analyses[14].

We performed two-sample Mendelian randomization using the MRBase tool with default settings[15]. We created a genetic instrument variable for CAD using the same GWAS as was used in the genetic correlation analysis[2]. We selected genome-wide significant SNPs that were determined to be independent using a clumping distance of 10 megabases with a linkage disequilibrium $R^2$ threshold of 0.001. These were then harmonized to the summary statistics of each outcome, excluding palindromic SNPs and using proxies for missing SNPs only if the LD $R^2$ was $\geq 0.9$. For lung cancer, the instrument variable consisted of 125 SNPs, of which 1 SNP was proxied. For breast cancer, the instrument variable consisted of 124 SNPs, of which none were proxied. MRBase was used to perform inverse variance weighted, weighted median, and MR Egger studies. As additional sensitivity analysis, and to test for horizonal pleiotropy, we used MR PRESSO[16].

WHI analyses were performed using SAS 9.4 (SAS Enterprise). UK Biobank analyses, meta-analysis, Mendelian randomization, and plots were done with R version 3.5.1 (R Foundation, Vienna, Austria). All odds ratios (OR) and hazard ratios (HR) are reported as per standard deviation increase in the PRS.

**Ethics statement**. The WHI project was reviewed and approved by the Fred Hutchinson Cancer Research Center (Fred Hutch) IRB in accordance with the U.S. Department of Health and Human Services regulations at 45 CFR 46 (approval number: IR# 3467-EXT). Participants provided written informed consent to participate. Additional consent to review medical records was obtained through signed written consent. Fred Hutch has an approved FWA on file with the Office for Human Research Protections (OHRP) under assurance number 0001920. WHI data were accessed through the sponsorship of T. Assimes (WHI co-investigator) and with an approved proposal (MSID 3914). The UK Biobank data was accessed under Application Number 13721. All participants gave informed consent for participation in UK Biobank. The Research Ethics Committee reference for UK Biobank is 16/NW/0274. This study of pre-existing de-identified data was deemed not human subjects research by the Stanford IRB, and thus no further consent was obtained.

**Reporting summary**. Further information on research design is available in the Nature Research Reporting Summary linked to this article.

## Results

We identified 25,789 subjects who had undergone genotyping as part of prior GWAS within the WHI (Supplementary Table 1). When plotting the first two principal components, we noted a cluster of 472 subjects from the GECCO study who were clear outliers (Supplementary Fig. 1A). The similar shape between the main cluster and the outliers suggested a batch effect leading to a systematic bias in genotyping calls. These subjects were removed. We then used the Mahalanobis distance[17] in the remaining subjects to identify a central cluster with similar genetically inferred ancestry (Supplementary Fig. 1B). The majority of these subjects self-reported as non-Hispanic white. Lastly, we excluded 2,830 subjects (11%) with known or likely ASCVD at enrollment (Supplementary Table 2, Supplementary Fig. 2). The remaining cohort of 21,863 subjects showed an enrichment for health traits and outcomes reflective of the genotyping strategy of the parent WHI GWAS, which targeted genotyping for outcomes of interest (Supplementary Table 3). Polygenic risk for CAD was quantified in each of these participants using a validated genome-wide PRS for CAD[5].

**Association of polygenic risk for CAD and quantitative measurements**. We identified 454 blood-based laboratory biomarkers for assessment with PRS after excluding biomarkers with fewer than 100 observations. Lab biomarkers were categorized as being related to lipids ($n = 84$), diabetes ($n = 7$), hormones ($n = 93$), inflammation ($n = 62$), hematology ($n = 38$), or other ($n = 170$). We further identified 48 clinical exam measurements, 31 quantitative traits reported by questionnaire, and 13 clinical scores. Associations with lipid-related labs, diabetes-related labs, and with blood pressure measurements were adjusted for cholesterol medication use, diabetes medication use, and blood pressure medication use respectively.

Polygenic risk for CAD associated with traits related to traditional risk factors and the metabolic/insulin resistance syndrome. For example, women with a higher PRS tended to have higher systolic blood pressure, larger waist-to-hip ratios, higher fasting insulin, higher LDL-C, higher triglycerides and lower HDL-C (Fig. 1, Supplementary Data). Subjects with a higher PRS also reported less healthy diets. Among lipid measurements, lipoprotein(a) [Lp(a)] showed the most significant association with the CAD PRS. This observation may reflect the very high genetic heritability of Lp(a) levels[18]. Across multiple lab categories, we observed associations with biomarkers known or hypothesized to relate to CAD risk, including sex hormone binding globulin[19], leptin[20], hematocrit[21], and hepatocyte growth factor[22]. We also observed a negative association with height, corroborating prior reports that genetically determined shorter stature is associated with a higher risk for CAD[23]. Analysis of questionnaire data demonstrated that women with higher polygenic risk for CAD tended to report a younger age of their father's and/or mother's death, and they reported experiencing menopause at a younger age. Interestingly, higher polygenic risk

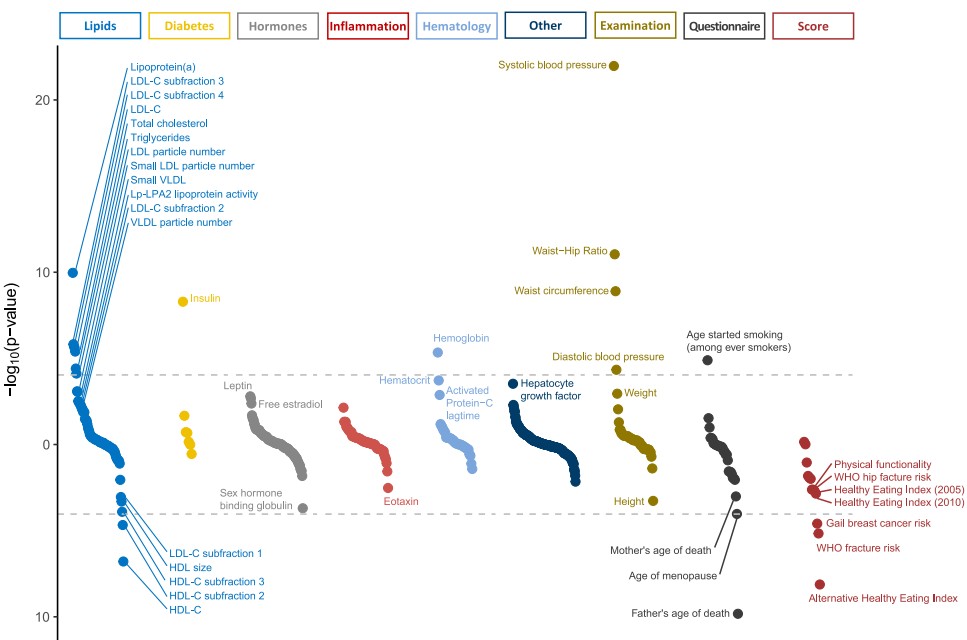

**Fig. 1 Associations between polygenic risk for coronary artery disease and quantitative traits derived from lab values, clinical exam, self-report, and clinical scores in the Women's Health Initiative.** Positive associations are plotted in the up direction and negative associations are plotted in the down direction. Associations that are significant with FDR $q$-value $\leq 0.05$ are labeled. The horizontal gray dashed lines represent the Bonferroni significant $p$-value $< 9.2 \times 10^{-5}$ (0.05/546). Sample size for each test is included in the Quantitative outcomes section of the Supplementary Data.

for CAD was associated with lower clinically predicted risk for breast cancer using the Gail breast cancer risk model[24].

**Association of polygenic risk for CAD and self-reports.** We aggregated data from structured questionnaires administered at baseline and during regular annual follow up and measured the association between polygenic risk for CAD and social/behavioral history, family history, medication usage, and self-reported medical history present at baseline or reported during follow-up. We compared three analyses in order to better understand the manifestations of polygenic risk for CAD in different contexts. The first analysis measured associations in the main study cohort; the second analysis measured associations in the main study cohort with an added adjustment for whether the participant had developed CAD at last follow-up; the third analysis measured associations among the subset of women with no CAD at last follow-up. Figure 2 shows those outcomes which are significant based on a FDR $q$-value $\leq 0.05$ in any of the three analyses. The complete results are shared in the Supplementary Data.

We observed associations between increased polygenic risk for CAD and known risk factors for CAD, in all three analyses. Among women free of CAD, a higher PRS was associated with a higher likelihood of reporting hypertension (OR 1.2, 95% CI 1.16-1.24) hypercholesterolemia (OR 1.17, 95% CI 1.12–1.23), rheumatoid arthritis (OR 1.11, 95% CI 1.03–1.19), and family history of myocardial infarction (OR 1.16, 95% CI 1.13–1.20) or stroke (OR 1.07, 95% CI 1.04–1.11). We also observed an interesting association with smoking. In all three analyses, subjects with increased polygenic risk for CAD were slightly less likely to have ever smoked. However, among women who reported having ever smoked, a higher PRS was associated with a higher likelihood of being a current smoker (Fig. 2). Possibly related to the association with continued smoking into later adulthood, subjects with increased polygenic risk for CAD were more likely to report a diagnosis of emphysema. Beyond known risk factors, we saw evidence that the genetic drivers of CAD risk may also impact risk for other diseases, including venous thromboembolism (VTE), thyroid disease, and gallbladder-related disease.

We detected an inverse association of the CAD PRS with self-reported cancer-related outcomes. Women with higher polygenic risk for CAD were less likely to report a history of breast cancer (OR 0.81, 95% CI 0.69–0.95) or non-melanoma skin cancer (OR 0.93, 95% CI 0.89-0.98). They were also less likely to report family history of colon cancer (OR 0.95, 95% CI 0.91–0.99), which may in part explain their lower likelihood of having ever undergone a colonoscopy (OR 0.96, 95% CI 0.93-0.99). These associations did not attenuate when adjusting for CAD or when analyzing the subset of CAD-free women (Fig. 2).

**Association of polygenic risk for CAD and incident cardio-vascular diseases.** We next aimed to measure the impact of polygenic risk for CAD on incident cardiovascular disease, independent of traditional risk factors. Using high-quality adjudicated outcomes, we examined the various manifestations of CAD, non-CAD atherosclerotic cardiovascular disease, and other non-atherosclerotic cardiovascular disease. In total, we considered 17 cardiovascular outcomes with at least 100 incident cases among our study cohort. These outcomes represent first-presentation incident events. We adjusted for smoking status, diabetes, systolic blood pressure, LDL-C, and HDL-C. As expected, outcomes related to CAD showed the strongest associations with the PRS. The more severe manifestations of CAD including myocardial infarction and the need for coronary revascularization demonstrated the largest effect sizes (Fig. 3). A similarly strong association was seen for the first presentation of hospitalized angina ("All angina"). However, the majority of such cases were treated with coronary revascularization. Angina without revascularization demonstrated a comparably weak association that did not reach nominal statistical significance. Stroke also demonstrated a clear association with polygenic risk for CAD, though with weaker effect sizes compared to CAD-related outcomes. The association with stroke was driven by the ischemic subtype. We observed no association observed with

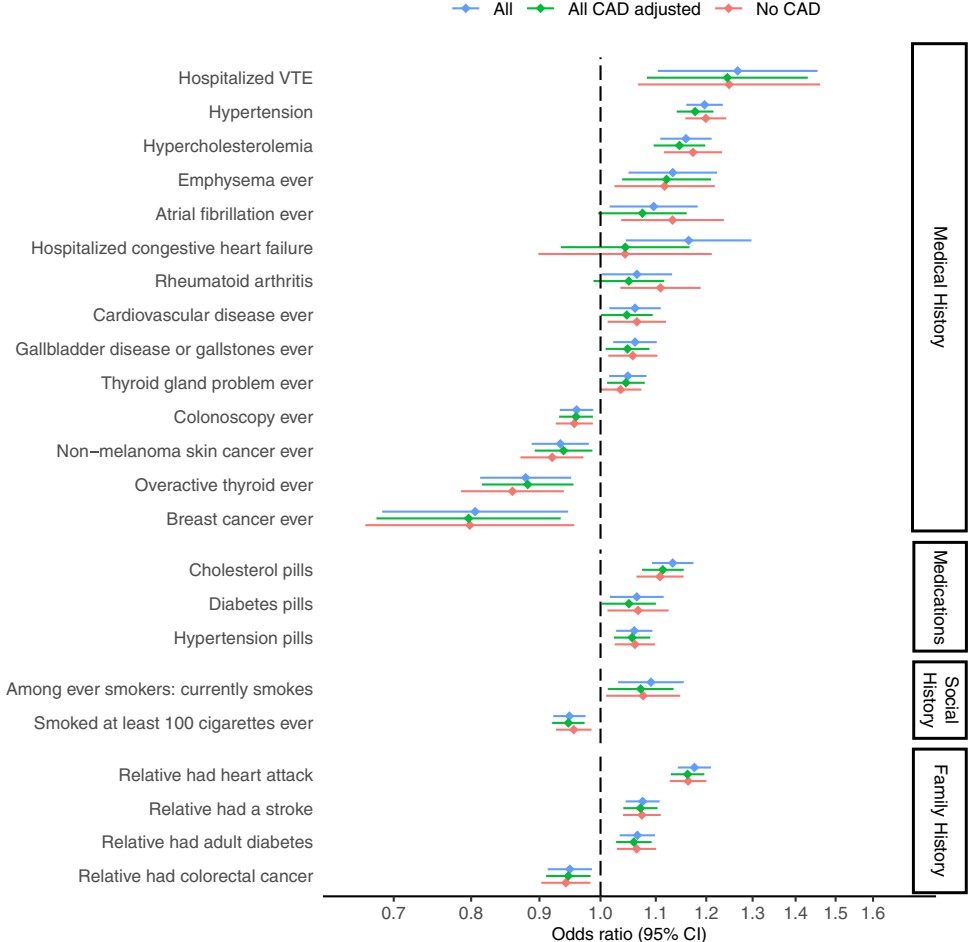

**Fig. 2 Associations between polygenic risk for coronary artery disease (CAD) and self-reported history, collected at baseline and throughout follow-up in the Women's Health Initiative.** Associations for three analyses are compared. 'All' shows associations in the main study cohort. 'All CAD adjusted' shows associations in the main study cohort with adjustment for presence/absence of CAD at last follow-up. 'No CAD' shows associations among the subset of participants with no CAD at last follow-up. Only outcomes with at least 100 cases in the CAD-free group were considered (128 outcomes). The plot shows all outcomes that were significant with FDR q-value ≤ 0.05 in any of the three analyses. Error bars represent 95% confidence intervals. The number of cases and controls for each test is included in the Self-reported outcomes section of the Supplementary Data.

hemorrhagic stroke. The significant association previously seen with VTE in the self-reported prevalent outcomes (Fig. 2) was not reflected in the adjudicated incident outcomes for pulmonary embolus of deep vein thrombosis (Fig. 3).

Results for a minimally adjusted model (adjusted only for age and genotype platform) are shown in Supplementary Table 4 for comparison. Consistent with prior studies[5], adjusting for ASCVD risk factors only slightly attenuates the strength of the PRS associations.

**Association of polygenic risk for CAD and incident cancers.** Results from our analyses of clinical scores and self-reported outcomes suggested the possibility of a protective association between polygenic risk for CAD and cancer. To better explore this finding, we tested the association between the CAD PRS and adjudicated first-occurrence incident cancers. We tested 17 cancers for which at least 100 incident cases occurred in our cohort. We found suggestive protective associations with the aggregate outcome of any cancer (OR 0.96, 95% CI 0.93–0.99, p 0.008) and with the specific outcomes of lung cancer (OR 0.91, 95% CI 0.84–0.99, p 0.02) and breast cancer (OR 0.96, 95% CI 0.93–1.00, p 0.05). Other cancers also showed a trend for the OR being less than 1 (Supplementary Fig. 3). After adjusting for cancer risk factors (smoking status, alcohol consumption, weekly physical

activity, dietary health measured by the alternative healthy eating index, and BMI), the associations with any cancer (OR 0.96, 95% CI 0.93–0.99, p 0.02), lung cancer (OR 0.91, 95% CI 0.83–0.99, p 0.02), and breast cancer (0.96, 95% CI 0.92–1.00, p 0.04) were virtually unchanged. We also tested for genotyping batch effects by repeating the analysis in the subset of women who were genotyped with the Oncochip. We found consistent effect sizes. Though, given substantially decreased sample sizes, the p-values were no longer significant (Supplementary Table 5).

We next sought to test whether the associations with breast cancer and lung cancer replicate in an external cohort. We used data from the UK Biobank to test the association between the CAD PRS and incident first-occurrence breast cancer or lung cancer among post-menopausal women with no history of MI or stroke at baseline. The association replicated for breast cancer but not for lung cancer. Using a random effects meta-analysis, only the breast cancer association remained significant (Fig. 4).

The observed inverse association between the CAD PRS and cancer outcomes may reflect several factors. One possibility is that CAD and cancer have some shared genetic architecture but with opposing effects. To test this hypothesis, we performed genetic correlations using linkage disequilibrium score regression with summary statistics of previously published GWAS for each outcome. We found a small but significant negative genetic

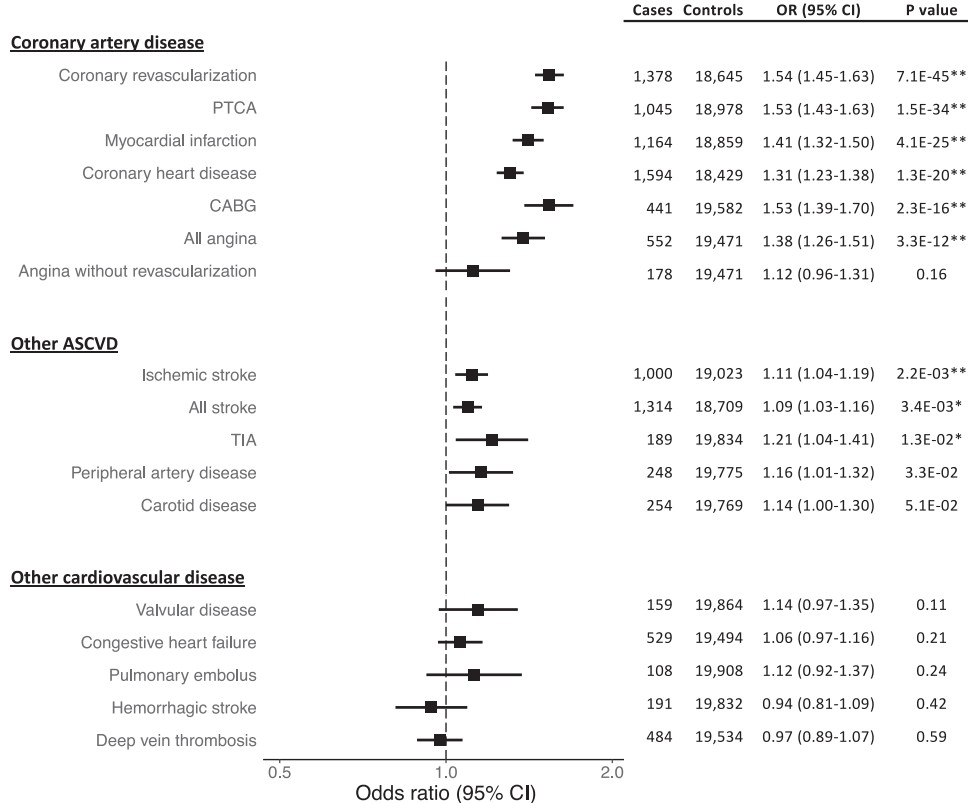

**Fig. 3 Associations between polygenic risk for coronary artery disease and incident adjudicated cardiovascular diseases in the Women's Health Initiative.** Outcomes with at least 100 incident cases were considered. Models were adjusted for smoking status, self-reported diabetes at baseline, systolic blood pressure, low-density lipoprotein cholesterol, and high-density lipoprotein cholesterol. Odds ratios (OR) are per 1 standard deviation increase in polygenic risk score. Error bars represent 95% confidence intervals. Outcomes with a single asterisk are significant with an FDR q-value ≤ 0.05. Outcomes with double asterisks have a Bonferroni significant p-value 0.003 (0.05/17). PTCA = Percutaneous Transluminal Coronary Angioplasty; CABG = Coronary Artery Bypass Graft; ASCVD = Atherosclerotic Cardiovascular Disease; TIA = Transient Ischemic Attack

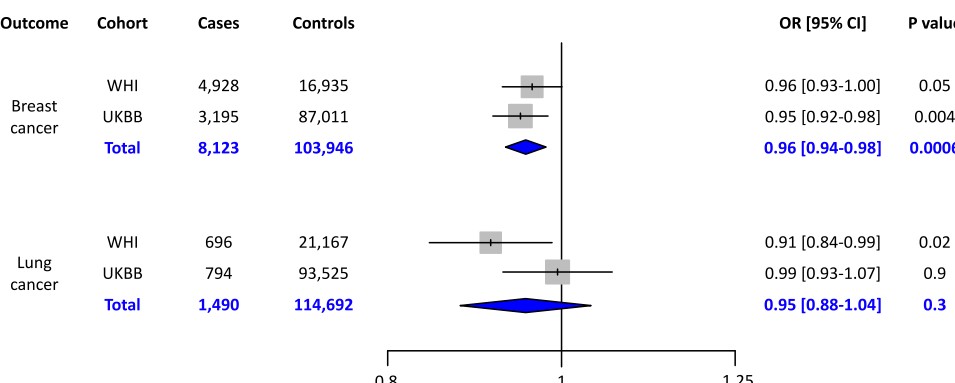

**Fig. 4 Associations between polygenic risk for coronary artery disease and incident breast and lung cancers in the Women's Health Initiative and the UK Biobank.** Random effects meta-analysis is shown in blue. Odds ratios (OR) are per 1 standard deviation increase in polygenic risk score. Error bars represent 95% confidence intervals. WHI = Women's Health Initiative; UKBB = UK Biobank

correlation between CAD and breast cancer (rg −0.054, p 0.02) but no significant genetic correlation between CAD and lung cancer (rg 0.053, p 0.4). We next tested the hypothesis that CAD is causally protective for breast and lung cancer using two-sample Mendelian randomization. Inverse variance weighted analysis of independent genome-wide significant SNPs suggested that CAD is causally protective for both breast cancer (OR 0.95, 95% CI 0.92–0.99, p 0.009) and lung cancer (OR 0.91, 95% CI 0.84–1.00, p 0.05). However, only the association with lung cancer was robust to sensitivity analyses with additional methods for Mendelian randomization, including weighted median, MR Egger[25], and MR

PRESSO[16]. Notably, the MR PRESSO global test detected significant horizontal pleiotropy for both the breast cancer and lung cancer analyses. After using the MR PRESSO correction for horizontal pleiotropy through outlier removal, the causal association between CAD and breast cancer was no longer significant, but the association between CAD and lung cancer remained significant (Table 1). Overall, these analyses suggest that the relationship between CAD and breast cancer is not causal, and the associations we observed in this study likely reflect shared genetic architecture. In contrast, the relationship between CAD and decreased risk for lung cancer may have a causal component.

**Table 1 Two-sample Mendelian randomization analyses testing for a causal association between the exposure of coronary artery disease and cancer outcomes.**

| Outcome | Method | OR (95% CI) | P Value |
|---|---|---|---|
| Breast cancer | Inverse variance weighted | 0.95 (0.92-0.99) | 0.009 |
| | Weighted median | 0.96 (0.92-1.01) | 0.1 |
| | MR Egger | 0.97 (0.90-1.04) | 0.4 |
| | MR PRESSO raw | 0.96 (0.93-1.00) | 0.04 |
| | MR PRESSO outlier-corrected | 0.98 (0.95-1.02) | 0.4 |
| Lung cancer | Inverse variance weighted | 0.92 (0.84-1.00) | 0.05 |
| | Weighted median | 0.84 (0.75-0.94) | 0.002 |
| | MR Egger | 0.77 (0.65-0.91) | 0.003 |
| | MR PRESSO raw | 0.91 (0.84-0.99) | 0.04 |
| | MR PRESSO outlier-corrected | 0.92 (0.86-0.99) | 0.04 |

| | | Cases | Non-cases | HR (95% CI) | P value |
|---|---|---|---|---|---|
| Coronary heart disease | | 382 | 19,641 | 1.29 (1.16-1.43) | 4.4E-06 |
| Unknown | | 130 | 19,893 | 1.28 (1.07-1.54) | 7.9E-03 |
| Cerebrovascular | | 651 | 19,372 | 1.11 (1.03-1.20) | 8.5E-03 |
| Dementia | | 587 | 19,436 | 1.11 (1.02-1.21) | 0.01 |
| Brain cancer | | 34 | 19,989 | 0.71 (0.52-0.97) | 0.03 |
| Pneumonia | | 255 | 19,768 | 1.14 (1.00-1.30) | 0.04 |
| Uterine cancer | | 22 | 20,001 | 0.68 (0.46-1.00) | 0.05 |

Hazard ratio (95% CI)

**Fig. 5 Association between polygenic risk for coronary artery disease and causes of death in the Women's Health Initiative.** Cox analysis was performed to measure the risk of death for each given cause. Non-cases were censored at time of death from any other cause or time of last follow-up. Models were adjusted for smoking status, self-reported diabetes at baseline, systolic blood pressure, low-density lipoprotein cholesterol, and high-density lipoprotein cholesterol. Hazard ratios (HR) are per 1 standard deviation increase in the polygenic risk score. Error bars represent 95% confidence intervals.

**Association of polygenic risk for CAD and causes of death**. We used a time-to-event analysis, accounting for competing events, to determine the impact of polygenic risk for CAD on cause of death. In total, 11,734 women died during the follow up period, with 78 distinct causes adjudicated. We measured the association between the CAD PRS and 48 causes of death for which at least 10 cases occurred among women with sufficient data to adjust for cardiovascular disease risk factors (smoking status, self-reported diabetes at baseline, systolic blood pressure, LDL-C, and HDL-C). Figure 5 shows all death outcomes that showed nominal significance. The complete results are available in the Supplementary Data. The strongest association occurred with 'definite' coronary heart disease death (HR 1.29, 95% CI 1.16–1.43). Conversely, there was no association with the outcome of 'possible' coronary heart disease death (HR 0.99, 95% CI 0.91–1.07), suggesting low specificity of that outcome. The magnitude of the association with unknown cause of death suggests that many of these deaths may have been secondary to ASCVD. Despite the observed inverse association with incident lung cancer and breast cancer, we did not find lower risk of death from either cancer. For lung cancer death, the HR was 0.93 (95% CI 0.85–1.02, p 0.14). For breast cancer death, the HR was 1.09 (95% CI 0.99–1.19, p 0.06). However, we did appreciate a nominally significant decreased risk for death from brain cancer and uterine cancer (Fig. 5), and similar to our analysis of incident cancer, we observed a trend for HR < 1 for cancer deaths (Supplementary Data).

**Discussion**

We have shown that polygenic risk for CAD, as quantified by a genome-wide PRS, has broad clinical manifestations in postmenopausal women. In addition to the expected association with CAD and other ASCVD outcomes, we observed associations with biomarkers, clinical measurements, behaviors, and disorders that are known to be risk factors for atherosclerosis. Recent work demonstrated an association between a 300-variant CAD PRS and traditional risk factors among participants of the UK Biobank[6]. Our analyses corroborate those findings and expand on them substantially by leveraging a genome-wide PRS and an extensively phenotyped population that includes exquisite adjudication for multiple outcomes in the setting of prolonged follow up.

Beyond traditional risk factors, we find that polygenic risk for CAD associates with risk-enhancing features that are defined in ASCVD prevention guidelines[26], including central adiposity, elevated Lp(a), and rheumatoid arthritis. We also highlight the heritable nature of polygenic risk through clear associations with early age of parental death as well as family history of MI and stroke. Notably, we find that polygenic risk for CAD associates with behaviors often referred to as "lifestyle risk factors", including dietary health and persistent smoking. Some consider lifestyle risk as "environmental," but our findings indicate that genetics may influence behaviors that impact one's exposure to such risk factors. Although healthy lifestyle can help to mitigate polygenic risk for CAD[27,28], the fact that genetic risk might impact lifestyle raises additional questions for exploration. It has been shown that interaction effects between polygenic risk and behaviorally mediated environmental exposures can exist[29]. It is possible that as PRS become more complex, such "gene by environment" interactions could become more influential in score behavior. These interactions have implications on the estimation of risk as the effect of alleles predisposing to such risk-related behaviors can only be expressed when the environmental factor is present. Subjects possessing high risk variants but never exposed to the adverse environment may have misspecification of their risk.

We found the effect sizes per standard deviation of CAD PRS for the most severe incident manifestations of CAD (i.e. myocardial infarction and coronary revascularization) to be consistent with the published literature for the same CAD PRS in validation cohorts of European-ancestry men and women combined[5,30]. One recent study using a different PRS documented heterogeneity in effects sizes between the sexes but did not report or adjust for differences in the severity of disease at presentation among males and females[31]. Given the substantially lower effect sizes we observed for an angina-only presentation, it is possible that heterogeneity of a PRS between any two groups can be influenced by differences in the case-mix of the severity/type of CAD at presentation. Collectively, the data to date suggest that a large majority of the CAD loci incorporated into the PRS affect a woman's risk of presenting with CAD to the same degree as they do men, even if the average age of presentation may be up to a decade later for women. Importantly, our study offers the opportunity to identify associations that may be specific to women. For example, we found that a higher CAD PRS was associated with younger age of menopause. Observational data has shown that early menopause is a risk factor for CAD[32,33], but recent Mendelian randomization suggests that this relationship is not causal[34]. Thus, our observed association between polygenic risk for CAD and age of menopause likely reflects shared heritable risk factors between CAD and early menopause.

Somewhat unexpectedly, several associations suggest that increased polygenic risk for CAD decreases the risk for cancer. Women with higher PRS had a lower Gail breast cancer risk score, and they were less likely to report prevalent breast cancer, non-melanoma skin cancer, or a family history of colorectal cancer. These findings were further corroborated by our analysis of incident adjudicated cancers, where we observed protective associations between polygenic risk for CAD and incident breast and lung cancer. The protective association with incident breast cancer replicated in the UK Biobank, and recently others have also replicated the association in other biobanks[35]. Our genetic correlation and Mendelian randomization studies suggest that this relationship between CAD and breast cancer is not causal but rather driven by horizontal pleiotropy. An overlapping genetic architecture for CAD and cancer is further supported by other observations. For example, we observed an association with higher polygenic risk for CAD and shorter stature, consistent with prior reports that genetically taller stature is associated with a lower risk for CAD and a higher risk of cancer[23,36]. Despite pleiotropy, it is also plausible that CAD is causally protective against some cancers. Indeed, our Mendelian randomization studies of lung cancer consistently suggested a causal relationship. One mechanism could be that the clinical management of CAD leads to interventions that are protective for some cancers. Engagement with the healthcare system, behavior changes, and treatments (e.g. aspirin[37,38] or statin[39]) could all potentially protect against incident cancer.

An important limitation of our analysis is that our study population is predominantly white by self-report. To date, CAD PRS have been primarily developed from GWAS in people of European ancestry, and they have been optimized for application to European-ancestry cohorts. This shortcoming remains a barrier to more broadly studying polygenic risk for CAD in diverse populations. With recent efforts to improve representation in CAD GWAS[7], we hope that future PRS research will expand to address this limitation. A second limitation of this work is that most of our analyses are correlative are primarily hypothesis-generating and/or hypothesis-supporting. Additional work across independent cohorts and with larger sample sizes is needed to further understand the relationships between CAD and cancer outcomes.

In conclusion, polygenic risk for CAD is associated with a broad spectrum of phenotypes. Many of these associations likely reflect the complex pathophysiology of CAD risk, while others may reflect pleiotropic effects beyond CAD. In particular, our findings motivate further exploration of the overlap between CAD and cancer biology.

## Data availability

The summary-level source data for each figure is included directly in the figures and/or in the Supplementary Data file. Individual-level WHI data is available with an approved proposal and sponsorship of a WHI investigator (https://www.whi.org/). UK Biobank data is available with an approved research proposal (https://www.ukbiobank.ac.uk/).

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

## Acknowledgements

The authors thank the WHI (Women's Health Initiative) participants, clinical sites, investigators, and staff for their dedicated efforts. The WHI program is funded by the National Heart, Lung, and Blood Institute, National Institutes of Health, U.S. Department of Health and Human Services through contracts 75N92021D00001, 75N92021D00002, 75N92021D00003, 75N92021D00004, 75N92021D00005.

## Author contributions

S.L.C., M.P., and T.L.A. conceived and designed the study. S.L.C., M.P., J.L., and C.T. performed the analyses. A.H.S., S.L., C.K., and J.E. M, provided critical feedback. S.L.C. and T.L.A. drafted the manuscript. All authors approved of the final manuscript submission.

## Competing interests

The authors declare no competing interests.
