## [Peer Review File · Communications Medicine]

Reviewers' comments:

Reviewer #1 (Remarks to the Author):

Clarke, et al. leverage a portion of the WHI data set that has been genotyped and identify a range of phenotypes that are associated with a characterized PRS for coronary heart disease. They identify a range of phenotype associations, many of which would be expected for a PRS associated with an atherosclerotic phenotype. They also observe some unexpected associations, such as with breast cancer. The strengths of the manuscript include the use of a large, well-characterized cohort, use of a well-characterized PRS and appropriate methods for the question being asked. The manuscript is clearly written and easy to follow. The manuscript would be enhanced by following up on one or more of the novel/unexpected associations in the manuscript to better determine whether the association is valid and to understand the underlying nature of the association. I suggest that they do this with the breast cancer association. Validation and characterization can be done using publicly available data. My specific comments follow:

1. Methods: The data set comprises 6 studies (per Supplementary Table 1) that are enriched for different diseases and genotyped on different platforms. When using a PRS that comprises a very large number of SNPs, platform effects can result in associations, which is exacerbated when platform is confounded with phenotype. In the Methods (page 6), the authors note that methods are adjusted for "study type" (clinical trial vs. other). Is study-type different than the 6 GWAS studies comprising the data set used in the analyses? If so, I would suggest adjusting for the genotyping platform/substudy and seeing how the results are impacted.

2. The cancer associations are interesting and should be probed further. I would suggest doing the following:

a. As noted above, because breast cancer cases were largely genotyped on a single genotyping platform (Illumina Oncochip), that authors should verify that the association is not due to a batch effect, as described above. In addition, they should test for an association just using the subset of women from the "Breast Cancer Post GWAS study" to see whether the association persist among this subset genotyped on a common platform.

b. What is the basis for the association? Is there a potential "causal" association? Or does this represent pleiotropy of a subset SNPs? Or are there shared risk factors? To probe this further, the authors can use publicly available summary statistics from CAD and breast cancer GWAS and measure: 1) the genetic correlation between these traits; 2) using Mendelian Randomization (MR) methods to explore a causal framework.

c. In the discussion, the authors suggestion early menopause as a potential etiological mechanism, and MR methods that us GWAS for age of menarche could be used to test whether this is a shared mechanism for each disease.

3. Discussion: The association with smoking is interesting. The disease-causing effects of a genetic predisposition to smoking are largely due to the effects of cigarette smoke (an environmental factor). The authors should comment on how CHD risk estimation using the MetaGRS would be affected when applied to individuals who smoke (who have the environmental exposure) vs. individuals who do not smoke. Their results suggest that the MetaGRS would be expected to overestimate CHD risk among individuals who do not smoke.

Reviewer #2 (Remarks to the Author):

Please see the attached review report.

Review Report for ‘Broad Clinical Manifestations of Polygenic Risk for Coronary Artery Disease in the Women’s Health Initiative’

This work focused on phenotype-wide association analyses using more than 100 coronary artery disease (CAD) related traits and clinical outcomes and investigated how a CAD polygenic risk score, metaGRS, associates with each of them. The study was carried out in a subset of participants in Women’s Health Initiative study with genotype information. Several associations with metaGRS were discovered in this hypothesis-generating analysis. Overall speaking, the paper was well written and easy to follow. However, I found its scientific and clinical contribution is relatively thin given the level of analytical thoughtfulness in this work. Below please see some of my comments and questions to the authors.

1. Estimations and corresponding inferences of associations should be listed in Result Section when they are mentioned instead of plain and subjective language such as ‘high/strong association’ or ‘tend to report a younger age...’. Presenting an actual estimation and its 95% confidence interval will be more informative to the readers.
2. Statistical significance was not defined in Method Section. Also how multiple comparison was handled was not clearly stated either. It was only mentioned in the Tables and Figures that FDR q-values and Bonferroni corrected p-values were considered.
3. In Result Section of ‘Association of polygenic risk for CAD and self-reports’ on page 8, the part ‘To minimize the risk of ascertainment...’ should be moved to Method Section as it stated how the analyses were conducted with the attempt to avoid ascertainment bias. At the same time, I found it is hard to follow the statement on ascertainment bias in co-morbidity driven by the diagnosis of CAD, and why restricting the analyses to a sub-sample with no CAD at the most recent follow-up would solve this bias. It would be great if the authors can be more specific about what they meant by ascertainment bias. If the authors were referring to confounding by indication of CAD, using an inverse probability weighting approach may be a better approach to adjust for it.
4. What’s the clinical implication of associations between PRS and phenotypic traits such as physical activity, diet scores, smoking, and whether receiving a colonoscopy? These associations are very likely confounded by some other factors. It will be helpful for the readers if the authors can provide rationales on investigating these associations.
5. There are several outcomes that could be treated as time-to-event data, for example death. The authors only considered a cross-sectional death status at age 85 years. When analyzing the association of metaGRS and death

due to a particular cause, death due to other causes should be viewed as competing risk events. Treating death as a binary outcome seems like not the best option as it oversimplifies the relations among death due to various causes.

6. Association analyses on majority of phenotypes (except lab measurements) seemed to use the same set of adjusted variables as illustrated in Method Section. However, this almost one-fits-all confounding adjustment weakens the association findings and their clinical values. For example, it was shown in this work that metaGRS has a negative association with breast cancer history. However, one of breast cancer known risk factors is breast density which is also likely (partially) driven by genetics. The observed negative association could be driven by the negative association between breast density and breast cancer risk, and the positive association between breast density and metaGRS. Failures to account for these known factors for the outcomes/traits could make the findings on PRS-trait association in this work just a repetition of what has been discovered. A more tailored adjustment for each model would address this issue at least to some degree.

**Broad Clinical Manifestations of Polygenic Risk for Coronary Artery Disease in the
Women's Health Initiative
Clarke et al.**

Response to reviewer comments

Reviewer #1 (Remarks to the Author):

Clarke, et al. leverage a portion of the WHI data set that has been genotyped and identify a range of phenotypes that are associated with a characterized PRS for coronary heart disease. They identify a range of phenotype associations, many of which would be expected for a PRS associated with an atherosclerotic phenotype. They also observe some unexpected associations, such as with breast cancer. The strengths of the manuscript include the use of a large, well-characterized cohort, use of a well-characterized PRS and appropriate methods for the question being asked. The manuscript is clearly written and easy to follow. The manuscript would be enhanced by following up on one or more of the novel/unexpected associations in the manuscript to better determine whether the association is valid and to understand the underlying nature of the association. I suggest that they do this with the breast cancer association. Validation and characterization can be done using publicly available data. My specific comments follow:

1. Methods: The data set comprises 6 studies (per Supplementary Table 1) that are enriched for different diseases and genotyped on different platforms. When using a PRS that comprises a very large number of SNPs, platform effects can result in associations, which is exacerbated when platform is confounded with phenotype. In the Methods (page 6), the authors note that methods are adjusted for “study type” (clinical trial vs. other). Is study-type different than the 6 GWAS studies comprising the data set used in the analyses? If so, I would suggest adjusting for the genotyping platform/substudy and seeing how the results are impacted.

We thank you for pointing out this important shortcoming in our initial analyses. We agree that failing to adjust for genotyping platform has the potential to lead to spurious associations. Thus, we have now added an adjustment for genotyping platform to all analyses. Overall, we did not see a significant change in our results. There were some slight changes in p-values and/or effect sizes, but our key findings and the overall patterns remained the same. In our methods, we now state that this adjustment is included for all analyses.

“Each model was adjusted for age at enrollment (or age at time of measurement for lab values), study type (clinical trial versus observational study), and genotyping platform.”

2. The cancer associations are interesting and should be probed further. I would suggest doing the following:

a. As noted above, because breast cancer cases were largely genotyped on a single genotyping platform (Illumina Oncochip), that authors should verify that the association is not due to a batch effect, as described above. In addition, they should test for an association just using the subset of women from the “Breast Cancer Post GWAS study”

to see whether the association persist among this subset genotyped on a common platform.

Thank you for this suggestion. We have now run the analysis on the subset of women who were genotyped with the Oncochip. Among this subset, we continue to see the same protective odds ratios as we do in the full analysis, but given the substantially decreased sample size, the p-values are no longer significant (although still trending towards nominal significance). We've added the following to the Results, and the new supplementary table is copied below.

“We also tested for genotyping batch effects by repeating the analysis in the subset of women who were genotyped with the Oncochip. We found consistent effect sizes. Though, given substantially decreased sample sizes, the p-values were no longer significant (Supplementary Table 4).”

Supplementary Table 4. Sensitivity analysis of adjudicated cancer outcomes among the subset of women genotyped with the Oncochip.

Outcome	Cases	Controls	Odds ratio	LCI	UCI	P value
Any cancer	4150	2808	0.96	0.91	1.01	0.1
Breast cancer	3734	3224	0.97	0.92	1.02	0.2
Lung cancer	167	6791	0.92	0.78	1.07	0.3

Overall, we find the consistency in effect size reassuring. Further, we have now tested for replication using the UK Biobank, which we believe is an even stronger validation. We were able to replicate our association for breast cancer, and in fact, a recent preprint from another group finds the same association with breast cancer in separate biobanks.

To the results, we have added the following, and we copy the new figure below as well:

“We next sought to test whether the associations with breast cancer and lung cancer replicate in an external cohort. We used data from the UK Biobank to test the association between the CAD PRS and incident first-occurrence breast cancer or lung cancer among women with no history of MI or stroke at baseline. The association replicated for breast cancer but not for lung cancer. Using a random effects meta-analysis, only the breast cancer association was significant (Figure 4).”

Figure 4. Associations between polygenic risk for coronary artery disease and incident breast and lung cancers in the Women’s Health Initiative and the UK Biobank. Random effects meta-analysis is shown in blue. Odds ratios (OR) are per 1 standard deviation increase in polygenic risk score.

To the discussion, we added the following reference to the recent preprint:

“The protective association with incident breast cancer replicated in the UK Biobank, and recently others have also replicated the association in other biobanks.”³⁵

b. What is the basis for the association? Is there a potential “causal” association? Or does this represent pleiotropy of a subset SNPs? Or are there shared risk factors? To probe this further, the authors can use publicly available summary statistics from CAD and breast cancer GWAS and measure: 1) the genetic correlation between these traits; 2) using Mendelian Randomization (MR) methods to explore a causal framework.

We agree that this is an interesting question that is immediately raised by our observational analysis. Following your recommendations, we have now performed genetic correlation and Mendelian randomization studies. We have focused on breast cancer and lung cancer, using published GWAS summary statistics. We found a small but significant genetic correlation between CAD and breast cancer, and our MR analysis suggested the relationship between CAD and breast cancer is not causal but rather due to horizontal pleiotropy. In contrast, for lung cancer, we did not find any genetic correlation, and MR analysis consistent showed evidence of a causal relationship. We have now added these findings to the Results along with a new Table, and we have added to the Discussion.

New Results paragraph:

“The observed inverse association between the CAD PRS and cancer outcomes may reflect several factors. One possibility is that CAD and cancer have some shared genetic architecture but with opposing effects. To test this hypothesis, we performed genetic correlations using linkage disequilibrium score regression with summary statistics of previously published GWAS for each outcome. We found a small but significant negative genetic correlation between CAD and breast cancer ($r_g -0.054$, $p 0.02$) but no significant genetic correlation between CAD and lung cancer ($r_g 0.053$, $p 0.4$). We next tested the hypothesis that CAD is causally protective for breast and lung cancer using two-sample Mendelian randomization. Inverse variance weighted analysis of independent genome-wide significant SNPs suggested that CAD is causally protective for both breast cancer (OR 0.95, 95% CI 0.92-0.99, $p 0.009$) and lung cancer (OR 0.91, 95% CI 0.84-1.00, $p 0.05$). However, only the association with lung cancer was robust to sensitivity analyses with additional methods for Mendelian randomization, including weighted median, MR Egger,²⁵ and MR PRESSO¹⁶. Notably, the MR PRESSO global test detected significant horizontal pleiotropy for both the breast cancer and lung cancer analyses. After using the MR PRESSO correction for horizontal pleiotropy through outlier removal, the causal association between CAD and breast cancer was no longer significant, but the association between CAD and lung cancer remained significant (Table). Overall, these analyses suggest that the relationship between CAD and breast cancer is not causal, and the associations we observed in this study likely reflect shared genetic architecture. In contrast, the relationship between CAD and decreased risk for lung cancer may have a causal component.”

Table. Two-sample Mendelian randomization analyses testing for a causal association between the exposure of coronary artery disease and cancer outcomes.

Outcome	Method	OR (95% CI)	P Value
Breast cancer	Inverse variance weighted	0.95 (0.92-0.99)	0.009
	Weighted median	0.96 (0.92-1.01)	0.1
	MR Egger	0.97 (0.90-1.04)	0.4
	MR PRESSO raw	0.96 (0.93-1.00)	0.04
	MR PRESSO outlier-corrected	0.98 (0.95-1.02)	0.4
Lung cancer	Inverse variance weighted	0.92 (0.84-1.00)	0.05
	Weighted median	0.84 (0.75-0.94)	0.002
	MR Egger	0.77 (0.65-0.91)	0.003
	MR PRESSO raw	0.91 (0.84-0.99)	0.04
	MR PRESSO outlier-corrected	0.92 (0.86-0.99)	0.04

New Discussion:

“Our genetic correlation and Mendelian randomization studies suggest that this relationship between CAD and breast cancer is not causal but rather driven by horizontal pleiotropy. An overlapping genetic architecture for CAD and cancer is further supported by other observations. For example, we observed an association with higher polygenic risk for CAD and shorter stature, consistent with prior reports that genetically taller stature is associated with a lower risk for CAD and a higher risk of cancer.^{23,36} Despite pleiotropy, it is also plausible that CAD is causally protective against some cancers. Indeed, our Mendelian randomization studies of lung cancer consistently suggested a causal relationship. One mechanism could be that the clinical management of CAD leads to interventions that are protective for some cancers. Engagement with the healthcare system, behavior changes, and treatments (e.g. aspirin^{37,38} or statin³⁹) could all potentially protect against incident cancer.”

c. In the discussion, the authors suggestion early menopause as a potential etiological mechanism, and MR methods that use GWAS for age of menarche could be used to test whether this is a shared mechanism for each disease.

We assume the reviewer meant to state “GWAS of age of menopause” rather than menarche.

Since our initial submission of this manuscript, we have completed a separate study that specifically examines age of natural menopause and coronary artery disease. Using both one-sample and two-sample MR across several cohorts, we do not find evidence of a causal association between ANM and incident CAD after menopause, despite a strong observational association. This work is now in revision for another journal. However, a preprint is available here: <https://www.medrxiv.org/content/10.1101/2022.01.26.22269835v1>

Given these findings, we believe the association between CAD and breast cancer is not due to ANM being causal for both. Rather, we believe that there are shared heritable risk factors for both CAD and early menopause. Further, as we describe above, our MR analysis suggests that CAD is not causally protective for breast cancer but rather there exists this association due to horizontal pleiotropy.

We have updated the discussion to remove the prior discussion of early menopause as a possible causal link between CAD and breast cancer, and we changed our discussion of menopause and CAD as follows:

“Observational data has shown that early menopause is a risk factor for CAD,^{32,33} but recent Mendelian randomization suggests that this relationship is not causal.³⁴ Thus, our observed association between polygenic risk for CAD and age of menopause likely reflects shared heritable risk factors between CAD and early menopause.”

3. Discussion: The association with smoking is interesting. The disease-causing effects of a genetic predisposition to smoking are largely due to the effects of cigarette smoke (an environmental factor). The authors should comment on how CHD risk estimation using the MetaGRS would be affected when applied to individuals who smoke (who have the environmental exposure) vs. individuals who do not smoke. Their results suggest that the MetaGRS would be expected to overestimate CHD risk among individuals who do not smoke.

This is an interesting point worth adding to the discussion although we would like to emphasize up front that the complex topic of genetic risk prediction and its value in clinical practice is not the focus of our report. We agree that if some of the predictive risk alleles incorporated into metaGRS associate with CAD specifically through their association with an increased likelihood of persistent smoking, then among never smokers who are carriers of those alleles, the metaGRS would likely overestimate risk. The degree of overestimation would depend on the fractional contribution of these alleles to genetic risk overall within the metaGRS which currently remains unclear. Sorting out such conditional effects on risk prediction is challenging because it requires that CAD GWAS be redone stratified by various risk-factor levels (e.g. by smoking status) and ideally free of any selection biases. Such subgroup effects on the performance of the PRS even among subjects within a single ancestry have been documented by others not only for risk factors (PMID: 29874179) but also socio-economic status/education (PMID: 31999256, 35134953) and primary and secondary treatments of disease such as statins (PMID: 25748612, 28223407). We also observed a heterogeneous effects of the metaGRS on smoking related status in WHI given an elevated genetic risk was associated with a higher frequency of persistent smoking (i.e. current smokers) among women ever smokers in late life but a lower frequency of having ever smoked 100 cigarettes. The field is ripe for deeper exploration of gene/environment interactions and their influence on risk prediction, and we hope our work here may trigger such investigations. We have now added to the discussion the following:

“Notably, we find that polygenic risk for CAD associates with behaviors often referred to as “lifestyle risk factors”, including dietary health and persistent smoking. Some consider lifestyle risk as “environmental,” but our findings indicate that genetics may influence behaviors that impact one’s exposure to such risk factors. Although healthy lifestyle can help to mitigate polygenic risk for CAD,^{27,28} the fact that genetic risk might impact lifestyle raises additional questions for exploration. It has been shown that interaction effects between polygenic risk and behaviorally mediated environmental exposures can exist.²⁹ It is possible that as PRS become more complex, such “gene by environment” interactions could become more influential in score behavior. These interactions have implications on the estimation of risk as the effect of alleles predisposing to such risk-related behaviors can only be expressed when the environmental factor is present. Subjects possessing high risk variants but never exposed to the adverse environment may have misspecification of their risk.”

Reviewer #2 (Remarks to the Author):

This work focused on phenotype-wide association analyses using more than 100 coronary artery disease (CAD) related traits and clinical outcomes and investigated how a CAD polygenic risk score, metaGRS, associates with each of them. The study was carried out in a subset of participants in Women's Health Initiative study with genotype information. Several associations with metaGRS were discovered in this hypothesis-generating analysis. Overall speaking, the paper was well written and easy to follow. However, I found its scientific and clinical contribution is relatively thin given the level of analytical thoughtfulness in this work. Below please see some of my comments and questions to the Authors.

1. Estimations and corresponding inferences of associations should be listed in Result Section when they are mentioned instead of plain and subjective language such as 'high/strong association' or 'tend to report a younger age...'. Presenting an actual estimation and its 95% confidence interval will be more informative to the readers.

We thank you for the thorough review and comments.

We have two comments to make in response to this suggestion. First, depending on the number of findings, restating all point estimates and confidence intervals that are already accessible through figures and tables can lengthen a manuscript unnecessarily and serve as a distraction for some readers who are looking for qualitative summary of findings already accessible within tables and figures. Some journals specifically request that such information not be duplicated in the results section.

Second, related to our first comment, we realized from this feedback that we did not adequately emphasize the intent and nature of our analysis. In this work, we aim to leverage the phenotype-wide association study (PheWAS) framework and apply it to a longitudinal cohort with deep phenotyping (i.e. documentation and/or measurement of observable traits related to health and disease). This application is different from the typical/standard PheWAS that relies on electronic health record (EHR) diagnosis codes only for such phenotyping. The PheWAS approach is an unbiased hypothesis-generating method that has had great success in driving genomics research in new and interesting directions. Given this approach, it is not possible to explicitly report effect estimates and 95% CI for every association, and often, we are more interested in overall patterns and trends. For this reason, we refer readers to our main figures and table, have selectively used plain language to summarize trends, and have shared the full set of results in the Supplementary Data file, so that reader may look up specific numbers for associations of interest if not reported directly in the main text.

2. Statistical significance was not defined in Method Section. Also how multiple comparison was handled was not clearly stated either. It was only mentioned in the Tables and Figures that FDR q-values and Bonferroni corrected p-values were considered.

We agree that this important detail was not communicated clearly in the appropriate section. Thus, we have now added a paragraph to the Statistical Analysis section of the Methods that describes our evaluation of statistical significance.

“Using a phenome-wide association study framework, we consider statistical significance in three ways. Nominal significance is defined as a p -value ≤ 0.05 . Where indicated, we also identify associations that are significant by a false-discovery rate (FDR) q -value ≤ 0.05 , using the Benjamini and Hochberg method. Lastly, Bonferroni significance is defined as 0.05 divided by the number of association tests performed for the given analysis. For the association analysis of quantitative traits, Bonferroni significance was p -value $\leq 9.2 \times 10^{-5}$ (0.05/546). For the association analysis of incident cardiovascular diseases, Bonferroni significance was p -value ≤ 0.003 (0.05/17).”

3. In Result Section of 'Association of polygenic risk for CAD and self-reports' on page 8, the part 'To minimize the risk of ascertainment...' should be moved to Method Section as it stated how the analyses were conducted with the attempt to avoid ascertainment bias. At the same time, I found it is hard to follow the statement on ascertainment bias in co-morbidity driven by the diagnosis of CAD, and why restricting the analyses to a sub-sample with no CAD at the most recent follow-up would solve this bias. It would be great if the authors can be more specific about what they meant by ascertainment bias. If the authors were referring to confounding by indication of CAD, using an inverse probability weighting approach may be a better approach to adjust for it.

We thank you for catching this error. Ascertainment bias is an inappropriate description for what we intended to communicate. Our main goal in this analysis was to identify associations with prevalent self-reported outcomes that are not arising primarily downstream of a diagnosis of CAD. For example, it is possible that the PRS associates with self-reported risk factors because the diagnosis of risk factors is more likely to be made after a subject is diagnosed with CAD. However, we show that even among subjects with no known CAD at follow up, the PRS associates with risk factors, such as hypertension. The second motivation for this analysis is to distinguish associations with outcomes that occur directly downstream of CAD. For example, the association between the PRS and hospitalized congestive heart failure attenuates substantially in our study when examining the subjects with no CAD at follow up. By comparison, the association with atrial fibrillation remains robust. Thus, we believe the association with heart failure likely reflects a well-established downstream manifestation of CAD, while the association with atrial fibrillation is independent of any CAD.

Such concerns are in line with what others have observed when conducting similar studies and providing estimates of association prior to and after the exclusion of subjects with the condition related to the PRS (in this case CAD) (e.g., Ntalla et al. JACC 2021; PMID: 31196449)

To make the intention of this subgroup analysis clearer, we have removed the previous text regarding ascertainment bias, and we have replaced it with the following:

“In order to identify prevalent associations independent of the development of CAD after enrollment, we also performed the analysis in a subset of participants with no self-reported or adjudicated CAD at the most recent follow-up ($n = 18,044$).”

4. What's the clinical implication of associations between PRS and phenotypic traits such as physical activity, diet scores, smoking, and whether receiving a colonoscopy? These associations are very likely confounded by some other factors. It will be helpful for the readers if the authors can provide rationales on investigating these associations.

As we discuss above, we approached this analysis from the perspective of PheWAS. The PheWAS approach arose following the success of GWAS. Prior to GWAS, genetic studies were largely driven by candidate-gene analyses. GWAS allowed for an unbiased (ie genome-wide) approach, leading to new discoveries and broader understanding of genetics and disease. Similarly, the concept of PheWAS is to unbiasedly assess associations across available phenotypes without relying on a “candidate-phenotype” approach. Similar to GWAS, PheWAS has been a catalyst for new discovery and research. However, prior PheWAS work has relied on EHR data. In our analysis, we apply the approach to the extensive phenotyping performed as part of WHI. Thus, for example, we did not specifically choose to investigate the association with receiving a colonoscopy. As we note in the manuscript, this approach is hypothesis-generating. An explanation for the colonoscopy association could be that women with higher polygenic risk for CAD were less likely to report family members with colon cancer, which may decrease the likelihood of colonoscopy. We updated the text as follows to help clarify this hypothesis:

“They were also less likely to report family history of colon cancer (OR 0.95, 95% CI 0.91-0.99), which may in part explain their lower likelihood of having ever undergone a colonoscopy (OR 0.96, 95% CI 0.93-0.99).”

In terms of the associations with behaviors, we suspect that genome-wide PRS like metaGRS incorporate variants that influence risk for CAD through impacts on behavioral tendencies. For example, data from the Million Veteran Program show that smoking trajectories are heritable, and there is a genetic correlation between smoking trajectory and CAD (PMID: 33082346). Disentangling the relative influences of genetics, behaviors, and environmental exposures is a challenging area of future research. To highlight this fact better, we have added to the Discussion, as described above in our response to comment 3 from Reviewer #1.

5. There are several outcomes that could be treated as time-to-event data, for example death. The authors only considered a cross-sectional death status at age 85 years. When analyzing the association of metaGRS and death due to a particular cause, death due to other causes should be viewed as competing risk events. Treating death as a binary outcome seems like not the best option as it oversimplifies the relations among death due to various causes.

We thank you for this suggestion. We have now improved our death analysis by using a Cox model. For each cause of death, we censor non-cases at the time of death from another cause or at the time of last follow up if not deceased. We also now adjust for cardiovascular disease risk factors. We have removed our prior death analysis and replaced it with this new analysis, which is described under the “Association of polygenic risk for CAD and causes of death” section of Results. Our new figure is copied below:

Figure 5. Association between polygenic risk for coronary artery disease and causes of death in the Women's Health Initiative. Cox analysis was performed to measure the risk of death for each given cause. Non-cases were censored at time of death from any other cause or time of last follow-up. Models were adjusted for smoking status, self-reported diabetes at baseline, systolic blood pressure, low-density lipoprotein cholesterol, and high-density lipoprotein cholesterol. Hazard ratios (HR) are per 1 standard deviation increase in the polygenic risk score.

6. Association analyses on majority of phenotypes (except lab measurements) seemed to use the same set of adjusted variables as illustrated in Method Section. However, this almost one-size-fits-all confounding adjustment weakens the association findings and their clinical values. For example, it was shown in this work that metaGRS has a negative association with breast cancer history. However, one of breast cancer known risk factors is breast density which is also likely (partially) driven by genetics. The observed negative association could be driven by the negative association between breast density and breast cancer risk, and the positive association between breast density and metaGRS. Failures to account for these known factors for the outcomes/traits could make the findings on PRS-trait association in this work just a repetition of what has been discovered. A more tailored adjustment for each model would address this issue at least to some degree.

We initially did not explore outcome-specific models given our intention to use a PheWAS framework as discussed above. However, thanks to this comment, we realized that we could increase the impact of this work by extending beyond PheWAS for key outcomes. We felt that our exploration of high-quality adjudicated outcomes makes our analysis stand out compared to other studies. Therefore, we have now performed analyses with risk factor adjustment. For cardiovascular outcomes, we adjusted for known traditional risk factors, including smoking status, diabetes, blood pressure, and lipid levels. These results are now shown in **Figure 3**. We also apply these covariates to our Cox analyses of death (shown above in response to comment 5). Our risk factor adjusted analysis of cardiovascular outcomes is now described in the Results under the section, **Association of polygenic risk for CAD and incident cardiovascular diseases**, and the new figure is copied below:

Figure 3. Associations between polygenic risk for coronary artery disease and incident adjudicated cardiovascular diseases in the Women’s Health Initiative. Outcomes with at least 100 incident cases were considered. Models were adjusted for smoking status, self-reported diabetes at baseline, systolic blood pressure, low-density lipoprotein cholesterol, and high-density lipoprotein cholesterol. Odds ratios (OR) are per 1 standard deviation increase in polygenic risk score. Outcomes with a single asterisk are significant with an FDR q-value ≤ 0.05 . Outcomes with double asterisks have a Bonferroni significant p-value 0.003 (0.05/17). PTCA = Percutaneous Transluminal Coronary Angioplasty; CABG = Coronary Artery Bypass Graft; ASCVD = Atherosclerotic Cardiovascular Disease; TIA = Transient Ischemic Attack

For cancer outcomes, we report our results with and without risk factor adjustment. For cancer risk factors, we used smoking status, alcohol consumption, weekly physical activity, dietary health measured by the alternative healthy eating index, and BMI. These results are now described in the section, **Association of polygenic risk for CAD and incident cancers:**

“We found nominally significant protective associations with the aggregate outcome of any cancer (OR 0.96, 95% CI 0.93-0.99, p 0.008) and with the specific outcome of lung cancer (OR

0.91, 95% CI 0.84-0.99, p 0.02). For breast cancer, the association was borderline nominally significant (0.96, 95% CI 0.93-1.00, p 0.05). Other cancers did not have significant associations, but the OR showed a general trend for being less than 1 (**Supplementary Figure 3**). After adjusting for cancer risk factors (smoking status, alcohol consumption, weekly physical activity, dietary health measured by the alternative healthy eating index, and BMI), the associations with any cancer (OR 0.96, 95% CI 0.93-0.99, p 0.02), lung cancer (OR 0.91, 95% CI 0.83-0.99, p 0.02), and breast cancer (0.96, 95% CI 0.92-1.00, p 0.04) were virtually unchanged.”

Also, in response to suggestions from Reviewer 1, we have done additional follow-up analyses for our novel finding of an inverse association between the CAD PRS and cancer outcomes. The additional analyses include replication, genetic correlation, and two-sample Mendelian Randomization. From these additional analyses (please see response to comment 2 from Reviewer 1), we feel we can now provide a reasonable explanation for the breast cancer association that is supported by multiple lines of evidence. Mainly, there are genetic factors that simultaneously increase risk for CAD while decreasing risk for breast cancer. We think this novel observation will compel future research as the field of cardio-oncology is now rapidly growing.

Reviewers' comments:

Reviewer #1 (Remarks to the Author):

The authors have done a nice job responding to my questions.

Reviewer #2 (Remarks to the Author):

Please see the attached file for my comments.

Review Report for Revision of ‘Broad Clinical Manifestations of Polygenic Risk for Coronary Artery Disease in the Women’s Health Initiative’

This revision was substantially clearer and showing interesting results using an external cohort. I am thankful for the authors addressing my comments from previous review. While some issues were addressed in this revision, two main issues still need further clarification and justification. Please see below for my comments for the revision.

1. Please move the description on page 10 “In order to identify prevalent associations independent of the development of CAD after enrollment, we also performed the analysis in a subset of participants with no self-reported or adjudicated CAD at the most recent follow-up (n = 18,044). Only outcomes with at least 100 cases among the CAD-free cohort were considered, resulting in a total of 128 self-reported qualitative variables.” to Method section.
2. For the purpose of investigating associations of the PRS to outcomes independent of developing CAD, I suggest the authors do a full sample analysis adjusting for CAD status (e.g. a binary indicator of whether CAD was reported in any visits prior to the time when the outcome was reported or adjudicated). Solely restricting to a subset with no CAD does not fully capture the true associations “independent” of CAD development as this is a biased subset with characteristics (including PRS itself) different from the overall cohort. If the authors are truly interested in reporting these associations in CAD-free cohort, it would be less confusing if they state the goal straightforwardly instead of calling these results as associations independent of CAD development.
3. It’s not clear if the replication analysis using UKBiobank were using the same eligibility criteria as WHI, e.g. restricting to women of the same age window and post-menopausal; hence it was not clear if the findings, especially the association between PRS and breast cancer incidence which could be potentially confounded by menopausal status, were really validated in this external cohort. Please provide relevant details for the readers. Meanwhile, if the validation analysis in UKBiobank was not restricted to women post-menopausal, menopausal status needs to be adjusted for in the model for evaluating the association between PRS and breast cancer outcome.
4. Figure 4 showed the association of PRS to breast cancer in WHI was 0.95 (95% CI: 0.92-0.98, p-value=0.004); to lung cancer was 0.92 (95% CI: 0.85-1, p-value=0.04) which were different from what were described in the Result Section and Supplemental Figure 3. Please clarify why these results are different.

5. The findings on the trend of protective effect of CAD-PRS on incidence cancer is interesting. However, none of the findings, except perhaps the composite cancer outcome, reached a meaningful significance level after accounting for multiple comparisons. I would suggest removing the wording such as 'significance' or 'novel' for the associations with cancer outcomes throughout the manuscript.
6. Typo on page 17, 4th last sentence: 'cance'.

Response to reviewer comments

Reviewer #1 (Remarks to the Author):

The authors have done a nice job responding to my questions.

We thank the reviewer for their thoughtful review of our manuscript.

Reviewer #2 (Remarks to the Author):

This revision was substantially clearer and showing interesting results using an external cohort. I am thankful for the authors addressing my comments from previous review. While some issues were addressed in this revision, two main issues still need further clarification and justification. Please see below for my comments for the revision.

- 1. Please move the description on page 10 “In order to identify prevalent associations independent of the development of CAD after enrollment, we also performed the analysis in a subset of participants with no self-reported or adjudicated CAD at the most recent follow-up (n = 18,044). Only outcomes with at least 100 cases among the CAD-free cohort were considered, resulting in a total of 128 self-reported qualitative variables.” to Method section.**

We have moved this text to the methods and have added additional details based on our response to comment #2 below. The new text included in the methods is copied below in our response to comment #2.

- 2. For the purpose of investigating associations of the PRS to outcomes independent of developing CAD, I suggest the authors do a full sample analysis adjusting for CAD status (e.g. a binary indicator of whether CAD was reported in any visits prior to the time when the outcome was reported or adjudicated). Solely restricting to a subset with no CAD does not fully capture the true associations “independent” of CAD development as this is a biased subset with characteristics (including PRS itself) different from the overall cohort. If the authors are truly interested in reporting these associations in CAD-free cohort, it would be less confusing if they state the goal straightforwardly instead of calling these results as associations independent of CAD development.**

Thank you for raising this important point, and we agree that analysis of the subset of participants with no CAD at follow-up introduces a selection bias. Nonetheless, we believe some audiences are interested in this framework for the purpose of understanding the genetic pleiotropy captured by polygenic risk scores. In a separate manuscript in press, we have performed a similar analysis in the Million Veteran Program, and others have performed this type of analysis in the UK Biobank (PMID: 31196449). Thus we have opted to keep this analysis while also adding your suggested analysis in which we adjust for CAD in the full cohort. Further, we have removed the confusing text regarding associations being “independent” of CAD.

We have added the following text to the Methods:

“For the analysis of self-reported outcomes, we compared three associations. First, we performed logistic regression using the main study cohort, adjusting for age at enrollment, study type, and genotyping platform. Second, we added an additional binary covariate to adjust for presence or

absence of CAD at last follow-up. Third, we analyzed the subset of participants with no CAD at follow-up (n = 18,044), adjusting for age at enrollment, study type, and genotyping platform. CAD at follow-up was determined using both self-report and adjudicated outcomes. Only outcomes with at least 100 cases among the CAD-free cohort were considered, resulting in a total of 128 self-reported qualitative variables."

In addition, we have updated the Results as follows and have updated Figure 2:

*"We compared three analyses in order to better understand the manifestations of polygenic risk for CAD in different contexts. The first analysis measured associations in the main study cohort; the second analysis measured associations in the main study cohort with an added adjustment for whether the participant had developed CAD at last follow-up; the third analysis measured associations among the subset of women with no CAD at last follow-up. **Figure 2** shows those outcomes which are significant based on a FDR q-value ≤ 0.05 in any of the three analyses."*

Figure 2. Associations between polygenic risk for coronary artery disease (CAD) and self-reported history, collected at baseline and throughout follow-up in the Women’s Health Initiative.

Associations for three analyses are compared. ‘All’ shows associations in the main study cohort. ‘All CAD adjusted’ shows associations in the main study cohort with adjustment for presence/absence of CAD at last follow-up. ‘No CAD’ shows associations among the subset of participants with no CAD at last follow-up. Only outcomes with at least 100 cases in the CAD-free group were considered (128 outcomes). The plot shows all outcomes that were significant with FDR q-value ≤ 0.05 in any of the three analyses.

- It’s not clear if the replication analysis using UKBiobank were using the same eligibility criteria as WHI, e.g. restricting to women of the same age window and post-menopausal; hence it was not clear if the findings, especially the association between PRS and breast cancer incidence which could be potentially confounded by

menopausal status, were really validated in this external cohort. Please provide relevant details for the readers. Meanwhile, if the validation analysis in UKBiobank was not restricted to women post-menopausal, menopausal status needs to be adjusted for in the model for evaluating the association between PRS and breast cancer outcome.

We agree with this comment, and we have now updated our UKB analysis to only include post-menopausal women. The associations with breast and lung cancer are similar to what we observed previously for the larger cohort. The odds ratio for breast cancer is 0.95 (95% CI 0.92-0.98, p-value 0.004), and the odds ratio for lung cancer is 0.99 (95% CI 0.93-1.07, p-value 0.9). We have updated our meta-analysis to now use these results rather than the previous results. We have also made changes to the meta-analysis in response to comment #4 and include the updated figure in our response to that comment below.

- 4. Figure 4 showed the association of PRS to breast cancer in WHI was 0.95 (95% CI: 0.92-0.98, p-value=0.004); to lung cancer was 0.92 (95% CI: 0.85-1, p-value=0.04) which were different from what were described in the Result Section and Supplemental Figure 3. Please clarify why these results are different.**

Thank you for catching this error. The values reported in the text are correct, but the figure was incorrect. In our previous revision, we had mistakenly copied into the manuscript a figure of the meta-analysis that was created prior to incorporating Reviewer 1’s request to adjust for genotyping platform in all analyses (i.e. these were the WHI results that we had reported in our first submission, meta-analyzed with UKB). Thus, all the results reported in the text were correctly updated but the figure was erroneous. We have now updated our figure such that the correct WHI results are meta-analyzed, and we have updated the UKB results to address comment #3 above.

Corrected Figure 4:

- 5. The findings on the trend of protective effect of CAD-PRS on incidence cancer is interesting. However, none of the findings, except perhaps the composite cancer outcome, reached a meaningful significance level after accounting for multiple comparisons. I would suggest removing the wording such as ‘significance’ or ‘novel’ for the associations with cancer outcomes throughout the manuscript.**

Thank you for this suggestion, which we think is very reasonable. We have carefully gone through the manuscript and made minor edits to change the language surrounding our cancer associations, removing the terms significant and novel. For example, we have updated the results section related to cancer to read as follows:

*"We found suggestive protective associations with the aggregate outcome of any cancer (OR 0.96, 95% CI 0.93-0.99, p 0.008) and with the specific outcomes of lung cancer (OR 0.91, 95% CI 0.84-0.99, p 0.02) and breast cancer (OR 0.96, 95% CI 0.93-1.00, p 0.05). Other cancers also showed a trend for the OR being less than 1 (**Supplementary Figure 3**)."*

6. Typo on page 17, 4th last sentence: 'cance'.

Thank you for catching this typo, which we have now fixed.

REVIEWERS' COMMENTS:

Reviewer #2 (Remarks to the Author):

The authors have addressed all of my comments in this revision. I am satisfied except one minor comment. Please add '(UKBB)' after UK Biobank in the caption of Figure 4 to clarify the use of the acronym. Thanks for the great work.

Response to Reviewers

Reviewer #2 (Remarks to the Author):

The authors have addressed all of my comments in this revision. I am satisfied except one minor comment. Please add '(UKBB)' after UK Biobank in the caption of Figure 4 to clarify the use of the acronym. Thanks for the great work.

We thank you for your helpful review of this manuscript. We have updated the figure legend appropriately to clarify the UKBB acronym.